# MESHLRM: LARGE RECONSTRUCTION MODEL FOR HIGH-QUALITY MESHES

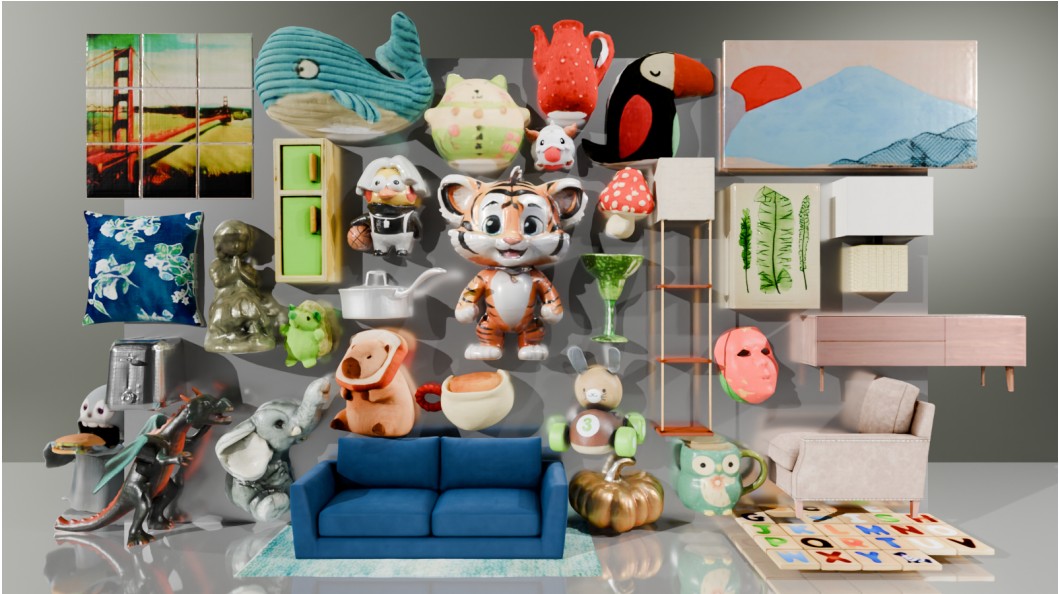

Figure 1: MeshLRM is an LRM-based content creation framework designed to produce high-quality 3D assets. The intricate 3D meshes and textures featured in this scene are all generated using our method in less than 1 second per asset, including an image/text-to-3D generation step and our end-to-end sparse-view mesh reconstruction.

## ABSTRACT

We propose MeshLRM, a novel LRM-based approach that can reconstruct a high-quality mesh from merely four input images in less than one second. Different from previous large reconstruction models (LRMs) that focus on NeRF-based reconstruction, MeshLRM incorporates differentiable mesh extraction and rendering within the LRM framework. This allows for end-to-end mesh reconstruction by fine-tuning a pre-trained NeRF LRM using mesh rendering. Moreover, we improve the LRM architecture by simplifying several complex designs in previous LRMs. MeshLRM's NeRF initialization is sequentially trained with low- and high-resolution images; this new LRM training strategy enables significantly faster convergence and thereby leads to better quality with less compute. Our approach achieves state-of-the-art mesh reconstruction from sparse-view inputs and also allows for many downstream applications, including text-to-3D and single-image-to-3D generation. https://meshlrm.github.io/

## 1 INTRODUCTION

High-quality 3D mesh models are the core of 3D vision and graphics applications that are specifically optimized for them, such as 3D editing, rendering, or simulation tools. 3D mesh assets are usually created manually by expert artists or reconstructed from multi-view 2D images. Traditionally this has been done with complex photogrammetry systems (Snavely et al., 2006; Furukawa & Ponce, 2009; Schonberger & Frahm, 2016; Schönberger et al., 2016), though recent neural representations, such as NeRF (Mildenhall et al., 2020), offer a simpler end-to-end pipeline through

per-scene optimization. These neural representations are however typically volumetric (Mildenhall et al., 2020; Müller et al., 2022; Chen et al., 2022; Liu et al., 2020; Yariv et al., 2020; Wang et al., 2021a) and converting them into meshes requires additional post-optimization (Tang et al., 2022; Yariv et al., 2023; Wei et al., 2023; Rakotosaona et al., 2023). Furthermore, both traditional and neural reconstruction approaches require a large number of input images and often long processing time (up to hours), limiting their interactivity.

Our goal is efficient and accurate 3D asset creation via sparse-view mesh reconstruction with direct feed-forward network inference and no post-optimization. We base our approach on recent large reconstruction models (LRMs) (Hong et al., 2024; Li et al., 2024; Xu et al., 2023; Wang et al., 2023a) for 3D reconstruction and generation. Existing LRMs use triplane NeRFs (Chan et al., 2022) as the 3D representation for high rendering quality. While these NeRFs can be converted into meshes via a Marching Cube (MC) (Lorensen & Cline, 1998) post-processing step. If done naively, this typically leads to a significant drop in rendering quality and geometric accuracy.

We propose to address this with MeshLRM, a novel transformer-based large reconstruction model, designed to directly output high-fidelity 3D meshes from sparse-view inputs. Specifically, we incorporate differentiable surface extraction and rendering into a NeRF-based LRM. We apply a recent Differentiable Marching Cube (DiffMC) (Wei et al., 2023) technique to extract an iso-surface from the triplane NeRF's density field and render the extracted mesh using a differentiable rasterizer (Laine et al., 2020). This lets us train MeshLRM end-to-end using a mesh rendering loss, optimizing it to produce density fields which are more compatible with the Marching Cube step, leading to more realistic and high-quality meshes.

We train MeshLRM by initializing the model using volumetric NeRF rendering; since our meshing components do not introduce any new parameters these pre-trained weights are a good starting point for our fine-tuning. We however find that fine-tuning a mesh-based LRM with differentiable MC (DiffMC) remains challenging. The primary challenge lies in the (spatially) sparse gradients from the DiffMC operation, that only affect surface voxels and leave the vast empty space untouched. This leads to poor local minima for model optimization and manifests as floaters in the reconstructions. We address this with a novel ray opacity loss that ensures that empty space along all pixel rays maintains near-zero density, effectively stabilizing the training and guiding the model to learn accurate floater-free surface geometry. Our end-to-end training approach reconstructs high-quality meshes with rendering quality that surpasses NeRF volumetric rendering (as shown in Tab. 5).

Our approach leverages a simple and efficient LRM architecture consisting of a large transformer model that directly processes concatenated multi-view image tokens and triplane tokens with self-attention layers to regress final triplane features that we use for NeRF and mesh reconstruction. In particular, we simplify many complex design choices used in previous LRMs (Li et al., 2024; Xu et al., 2023; Wang et al., 2023a), including the removal of pre-trained DINO modules in image tokenization and replacing the large triplane decoder MLP with small two-layer ones. We train MeshLRM on the Objaverse dataset (Deitke et al., 2023) with a progressive resolution strategy with low-resolution early training and high-resolution fine-tuning. These design choices lead to a state-of-the-art LRM model with faster training and inference and higher reconstruction quality.

In summary, our key contributions are:

- A LRM-based framework that integrates differentiable mesh extraction and rendering for end-to-end few-shot mesh reconstruction.
- A ray opacity loss for improved DiffMC-based training stability.
- A simplified LRM architecture and enhanced training strategies for fast and high-quality reconstruction.

We benchmark MeshLRM for 3D reconstruction (on synthetic and real datasets) and 3D generation (in combination with other multi-view generation methods). **Fig. 1 showcases high-quality mesh outputs from MeshLRM that were each reconstructed in under one second.**

## 2 RELATED WORK

**Mesh Reconstruction.** Despite the existence of various 3D representations, meshes remain the most widely used format in industrial 3D engines. Reconstructing high-quality meshes from multi-view

images is a long-standing challenge in computer vision and graphics. Classically, this is addressed via a complex multi-stage photogrammetry pipeline, integrating techniques such as structure from motion (SfM) (Snavely et al., 2006; Agarwal et al., 2011; Schonberger & Frahm, 2016), multi-view stereo (MVS) (Furukawa & Ponce, 2009; Schönberger et al., 2016), and mesh surface extraction (Lorensen & Cline, 1998; Kazhdan et al., 2006). Recently, deep learning-based methods have also been proposed to address these problems for higher efficiency and accuracy (Vijayanarasimhan et al., 2017; Tang & Tan, 2018; Yao et al., 2018; Huang et al., 2018; Im et al., 2019; Cheng et al., 2020). However, the photogrammetry pipeline requires dense input images and long processing time, aiming at exact object reconstruction, but often suffers from challenging calibration or appearance effects on the captures object. As opposed to this classic many-image acquisition process, our approach enables sparse-view mesh reconstruction through direct feed-forward network inference in an end-to-end manner.

**Neural Reconstruction.** Neural rendering techniques have recently gained significant attention for their ability to produce high-quality 3D reconstructions for realistic rendering (Zhou et al., 2018; Lombardi et al., 2019; Mildenhall et al., 2020; Kerbl et al., 2023). Most recent methods are based on NeRF (Mildenhall et al., 2020) and reconstruct scenes as various volumetric radiance fields (Mildenhall et al., 2020; Chen et al., 2022; Fridovich-Keil et al., 2022; Müller et al., 2022; Xu et al., 2022) with per-scene rendering optimization. Though radiance fields can be converted into meshes with Marching Cubes (Lorensen & Cline, 1998), the quality of the resulting mesh is not guaranteed. Previous methods have aimed to transform radiance fields into implicit surface representations (SDFs) (Wang et al., 2021a; Yariv et al., 2021; Oechsle et al., 2021), enhancing mesh quality from Marching Cubes. In addition, some methods (Yariv et al., 2023; Wei et al., 2023; Tang et al., 2022; Rakotosaona et al., 2023) attempt to directly extract meshes from radiance fields using differentiable surface rendering. However, these methods all require dense input images and long per-scene optimization time, which our approach avoids.

On the other hand, generalizable neural reconstruction has been explored, often leveraging MVS-based cost volumes (Chen et al., 2021; Long et al., 2022; Johari et al., 2022) or directly aggregating multi-view features along pixels' projective epipolar lines (Yu et al., 2021; Wang et al., 2021b; Suhail et al., 2022). While enabling fast and few-shot reconstruction, these methods can only handle small baselines, still requiring dense images with local reconstruction in overlapping windows to model a complete object. More recently, transformer-based large reconstruction models (LRMs) (Hong et al., 2024; Li et al., 2024; Wang et al., 2023a; Xu et al., 2023) have been proposed and achieved 3D NeRF reconstruction from highly sparse views. Inspired by this line of work, we propose a more efficient LRM architecture, incorporating DiffMC techniques (Wei et al., 2023) to enable high-quality sparse-view mesh reconstruction via direct feed-forward inference.

**3D Generation.** Generative models have seen rapid progress with GANs (Goodfellow et al., 2014) and, more recently, Diffusion Models (Sohl-Dickstein et al., 2015) for image and video generation. In the context of 3D generation, many approaches utilize 3D or multi-view data to train 3D generative models with GANs (Wu et al., 2016; Henzler et al., 2019; Gao et al., 2022; Chan et al., 2022) or diffusion models (Anciukevičius et al., 2023; Xu et al., 2023; Wang et al., 2023c; Chen et al., 2023a; Liu et al., 2023b), which are promising but limited by the scale and diversity of 3D data. DreamFusion (Poole et al., 2022) proposed to leverage the gradient of a pre-trained 2D diffusion model to optimize a NeRF for text-to-3D generation with a score distillation sampling (SDS) loss, achieving diverse generation results beyond common 3D data distributions. This approach inspired many follow-up approaches, to make the optimization faster or improve the results quality (Lin et al., 2023; Wang et al., 2023d; Tang et al., 2023; Chen et al., 2023b; Metzer et al., 2022). However, these SDS-based methods rely on NeRF-like per-scene optimization, which is highly time-consuming, often taking hours. More recently, a few attempts have been made to achieve fast 3D generation by utilizing pre-trained 2D diffusion models to generate multi-view images and then perform 3D reconstruction (Li et al., 2024; Liu et al., 2023a; 2024b; Tochilkin et al., 2024; Tang et al., 2024; Liu et al., 2024c). In particular, Instant3D (Li et al., 2024) leverages an LRM to reconstruct a triplane NeRF from four sparse generated images. In this work, we focus on the task of sparse-view reconstruction and propose to improve and re-target a NeRF LRM to directly generate meshes with higher quality. Unlike MeshGPT (Siddiqui et al., 2024), which sequentially predicts triangles using a transformer, we incorporate differentiable marching cubes and rasterization into the LRM framework to generate meshes. Concurrently, InstantMesh (Xu et al., 2024) makes a similar attempt by combining

Figure 2: MeshLRM architecture. The images are first patchified to tokens and fed to the transformer, concatenated with triplane positional embedding tokens. The output triplane tokens are unpatchified and decoded by two tiny MLPs for density and color. This model supports both the volumetric rendering (Sec. 3.2) and the DiffMC fine-tuning (Sec. 3.3).

OpenLRM He & Wang (2023) with Flexicubes Shen et al. (2023) for direct mesh generation. In contrast, we propose a novel simplified LRM architecture with a carefully designed training strategy and supervision losses, achieving superior quality over InstantMesh. Our model can be naturally coupled with a multi-view generator, such as the ones in (Li et al., 2024; Shi et al., 2023a), to enable high-quality text-to-3D and single-image-to-3D generation (see Fig. 7 and Fig. 6).

## 3 METHOD

We present MeshLRM enabling the reconstruction of high-quality meshes in under 1 second. We start with describing our backbone transformer architecture (Sec. 3.1), which simplifies and improves previous LRMs. We then introduce our two-stage framework for training the model: we first (Sec. 3.2) train the model to predict NeRF from sparse input images by supervising volume renderings at novel views, followed by refining the model for mesh surface extractions (Sec. 3.3) by performing differentiable marching cubes on the predicted density field and minimizing a surface rendering loss with differentiable rasterization.

### 3.1 MODEL ARCHITECTURE

As shown in Fig. 2, we propose a simple transformer-based architecture for MeshLRM, mainly consisting of a sequence of self-attention-based transformer blocks over concatenated image tokens and triplane tokens, similar to PF-LRM (Wang et al., 2023a). In contrast to PF-LRM and other LRMs (Hong et al., 2024; Li et al., 2024; Xu et al., 2023), we simplify the designs for both image tokenization and triplane NeRF decoding, leading to fast training and inference.

**Input posed image tokenization.** MeshLRM adopts a simple tokenizer for posed images, inspired by ViT (Dosovitskiy et al., 2020). We convert the camera parameters for each image into Plücker ray coordinates (Plücker, 1865) and concatenate them with the RGB pixels (3-channel) to form a 9-channel feature map. We then split the feature map into non-overlapping patches, and linearly transform them to input our transformer. With this process, the model does not need additional positional embedding as in ViT, since Plücker coordinates contain spatial information.

Note that our image tokenizer is much simpler than previous LRMs that use a pre-trained DINO ViT (Caron et al., 2021) for image encoding. We find that the pre-trained DINO ViT is unnecessary, possibly because it is mainly trained for intra-view semantic reasoning, while 3D reconstruction requires inter-view low-level correspondences. Vision transformer models (like DINO) often lack the zero-shot ability to handle untrained higher resolutions and require extensive fine-tuning to generalize (see Fig. 8). By dropping this per-view DINO encoding, the model becomes shallower, enabling a better connection between raw pixel information and 3D-related processing.

**Transformer.** We concatenate multi-view image tokens and learnable triplane (positional) embeddings, and feed them into a sequence of transformer blocks (Vaswani et al., 2017), where each block

is comprised of self-attention and MLP layers. We add layer normalization (Ba et al., 2016) before both layers (i.e., Pre-LN architecture), and use residual connections. This deep transformer network enables comprehensive information exchange among all the tokens, and effectively models intra-view, inter-view, and cross-modal relationships. The output triplane tokens, contextualized by all input views, are then decoded into the renderable triplane NeRF while the output image tokens are dropped. More specifically, each triplane token is unprojected with a linear layer and further unpatchified to $8 \times 8$ triplane features via reshaping. [1] All predicted triplane features are then assembled into the final triplane NeRF.

**Tiny density and color MLPs.** Previous LRMs (Hong et al., 2024; Li et al., 2024; Xu et al., 2023; Wang et al., 2023a) use a heavy shared MLP (e.g., 9 hidden layers with a hidden dimension of 64) to decode densities and colors from triplane features. This design leads to slow rendering in training. This is a bottleneck for MeshLRM since we need to compute a dense density grid, using the MLP and triplane features, for extracting the mesh with DiffMC. Therefore, we opt to use tiny MLPs with a narrower hidden dimension of 32 and fewer layers. In particular, we use an MLP with one hidden layer for density decoding and another MLP with two hidden layers for color decoding. Compared to a large MLP, our tiny MLPs lead to  50% speed-up in training without compromising quality (see Tab. 4). We use separate MLPs since the density MLP and color MLP are used separately in the Marching Cubes and surface rendering processes, respectively. We empirically find this MLP separation largely improves the optimization stability for DiffMC fine-tuning (described in Sec. 3.3), which is critical to large-scale training.

While being largely simplified, our transformer model can effectively transform input posed images into a triplane NeRF for density and color decoding. The density and color are then used to achieve both radiance field rendering for 1st-stage volume initialization (Sec. 3.2) and surface extraction & rendering for 2nd-stage mesh reconstruction (Sec. 3.3).

## 3.2 STAGE 1: EFFICIENT TRAINING FOR VOLUME RENDERING

We train our model with ray marching-based radiance field rendering Mildenhall et al. (2020) to bootstrap our model, providing good initialized weights for mesh reconstruction. Instead of training directly using high-res ($512 \times 512$) input images like prior LRM works (Hong et al., 2024; Li et al., 2024), we develop an efficient training scheme, inspired by ViT (Dosovitskiy et al., 2020). Specifically, we first pretrain our model with $256 \times 256$ input images until convergence and then finetune it for fewer iterations with $512 \times 512$ input images. This training schedule leads to significantly better quality than training at 512-res from scratch given the same amount of compute (see Tab. 1).

**256-res pretraining.** The pre-training uses 256-resolution images for both model input and output. We use a batch size of 8 objects per GPU and sample 128 points per ray during ray marching. In contrast to training with 512-res images from scratch (like previous LRMs), our training efficiency benefits from the low-res pre-training from two factors: shorter sequence length for computing self-attention and fewer samples per ray for volume rendering (compared to our high-res fine-tuning).

**512-res finetuning.** For high-res fine-tuning, we use 512-resolution images for the model input and output. We use a batch size of 2 per GPU and densely sample 512 points per ray. Here, we compensate for the increased computational costs (from longer token sequences and denser ray samples) by reducing the batch size 4 times, achieving a similar per iteration time to the low-res pretraining. It's also worth noting that our resolution-varying pretraining and finetuning scheme benefits from the use of Plücker coordinates for camera conditioning. It avoids the positional embedding interpolation (Dosovitskiy et al., 2020; Radford et al., 2021) and has inherent spatial smoothness.

**Loss.** We use an L2 regression loss $L_{v,r}$ and a perceptual loss $L_{v,p}$ (proposed in Chen & Koltun (2017)) to supervise the renderings from both phases. Since rendering full-res images is not affordable for volume rendering, we randomly sample a $128 \times 128$ patch from each target 256- or 512-res image for supervision with both losses. We also randomly sample 4096 pixels per target image for additional L2 supervision, allowing the model to capture global information beyond a single patch.

---

[1]This operator is logically the same to a shared 2D-deconvolution with stride 8 and kernel 8 on each plane.

The loss for volume rendering training $L_v$ is:

$$L_v = L_{v,r} + w_{v,p} * L_{v,p} \tag{1}$$

where we use $w_{v,p} = 0.5$ for both 256- and 512-res training.

### 3.3 STAGE 2: STABLE TRAINING FOR SURFACE RENDERING

Once trained with volume rendering, our model already achieves high-fidelity sparse-view NeRF reconstruction, which can be rendered with ray marching to create realistic images. However, directly extracting a mesh from the NeRF's density field results in significant artifacts as reflected by Tab. 2. Therefore, we propose to fine-tune the network using differentiable marching cubes and differentiable rasterization, enabling high-quality feed-forward mesh reconstruction.

**Mesh extraction and rendering.** We compute a $256^3$ density grid by decoding the triplane features and adopt a recent differentiable marching cubes (DiffMC) technique (Wei et al., 2023) to extract mesh surface from the grid. The DiffMC module is based on a highly optimized CUDA implementation, much faster than existing alternatives (Shen et al., 2023; 2021), enabling fast training and inference for mesh reconstruction.

To compute the rendering loss, we render the generated mesh using the differentiable rasterizer Nvdiffrast (Laine et al., 2020) and utilize the triplane features to (neurally) render novel images from our extracted meshes. This full rendering process is akin to deferred shading where we first obtain per-pixel XYZ locations via differentiable rasterization before querying the corresponding triplane features and regressing per-pixel colors using the color MLP. We supervise novel-view renderings with ground-truth images, optimizing the model for high-quality end-to-end mesh reconstruction.

However, using the rendering loss alone leads to high instability during training and severe floaters in the meshes (see Fig. 3). This is caused by the sparsity of density gradients in mesh rendering: unlike volume rendering which samples points throughout the entire view frustum, mesh rendering is restricted to surface points, lacking gradients in the empty scene space beyond the surface.

**Ray opacity loss.** To stabilize the training and prevent the formation of floaters, we propose to use a ray opacity loss. This loss is applied to each rendered pixel ray, expressed by:

$$L_\alpha = ||\alpha_{\mathbf{q}}||_1, \quad \alpha_{\mathbf{q}} = 1 - \exp(-\sigma_{\mathbf{q}}||\mathbf{p} - \mathbf{q}||) \tag{2}$$

where $\mathbf{p}$ represents the ground truth surface point along the pixel ray, $\mathbf{q}$ is randomly sampled along the ray between $\mathbf{p}$ and camera origin, and $\sigma_{\mathbf{q}}$ is the volume density at $\mathbf{q}$; when no surface exists for a pixel, we sample $\mathbf{q}$ inside the object bounding box and use the far ray-box intersection as $\mathbf{p}$.

In essence, this loss is designed to encourage the empty space in each view frustum to contain near-zero density. Here we minimize the opacity value $\alpha_{\mathbf{q}}$, computed using the ray distance from the sampled point to the surface. This density-to-opacity conversion functions as an effective mechanism for weighting the density supervision along the ray with lower loss values for points sampled closer to the surface. An alternative approach would be to directly regularize the density $\sigma_{\mathbf{q}}$ using an L1 loss, but we found this to lead to holes in surfaces as all points contribute the same, regardless of their distance to the surface. A more detailed comparison between our loss and the standard density and depth losses can be found in Appendix A.

**Combined losses.** To measure the visual difference between the results renderings after our DiffMC step and ground-truth (GT) images, we use an L2 loss $L_{m,r}$ and a perceptual loss $L_{m,p}$ similar to stage 1. We still supervise the opacity using $L_\alpha$, a ray opacity loss around the surface points using the GT depth maps. In addition, to further improve our geometry accuracy and smoothness, we apply an L2 normal loss $L_n$ to supervise the face normals of our extracted mesh with GT normal maps in foreground regions. Our final loss for mesh reconstruction is

$$L_m = L_{m,r} + w_{m,p} * L_{m,p} + w_\alpha * L_\alpha + w_n * L_n \tag{3}$$

where we use $w_{m,p} = 2$, $w_\alpha = 0.5$ and $w_n = 1$ in our experiments. Since mesh rasterization is significantly cheaper than volume ray marching, we render the complete images ($512 \times 512$ in our experiment) for supervision, instead of the random patches+rays used in Stage 1 volume training.

Table 1: Comparison between the model trained from scratch with 512 resolution images and the model trained with our pretraining + finetuning strategy. Note that the two models use the same compute budget — 128 A100 (40G VRAM) GPUs for 64 hours.

| | PSNR↑ | SSIM↑ | LPIPS↓ | #Pts/Ray | BatchSize | $T_{\text{iter}}$ | #Iter | $T_{\text{total}}$ |
|---|---|---|---|---|---|---|---|---|
| 512-res from scratch | 25.53 | 0.892 | 0.123 | 512 | 1024 | 7.2s | 32k | 64hrs |
| 256-res pretrain | **28.13** | **0.923** | **0.093** | 128 | 1024 | 2.6s | 60k | 44hrs |
| 512-res fine-tune | | | | 512 | 256 | 3.6s | 20k | 20hrs |

## 4 EXPERIMENTS

### 4.1 DATASET AND EVALUATION PROTOCOLS

Our model is trained on the Objaverse dataset (Deitke et al., 2023) (730K objects) for the 1st-stage volume rendering training and is subsequently fine-tuned on the Objaverse-LVIS subset (46K objects) for the 2nd-stage surface rendering finetuning. Empirically, the Objaverse-LVIS subset has higher quality and previous work (Rombach et al., 2022; Wei et al., 2021; Dai et al., 2023) shows that fine-tuning favors quality more than quantity. We evaluate the reconstruction quality of MeshLRM alongside other existing methods on the GSO (Downs et al., 2022), NeRF-Synthetic (Mildenhall et al., 2020), and OpenIllumination (Liu et al., 2024a) datasets, employing PSNR, SSIM, and LPIPS as metrics for rendering quality and bi-directional Chamfer distance (CD) as the metric for mesh geometry quality (more details in Appendix H).

### 4.2 ANALYSIS AND ABLATION STUDY

**Volume rendering (Stage 1) training strategies.** To verify the effectiveness of our training strategy that uses 256-res pretraining and 512-res fine-tuning (details in Sec. 3.2), we compare with a model having the same architecture but trained with high-resolution only (i.e., 512-res from scratch). Tab. 1 shows the quantitative results on the GSO dataset with detailed training settings and timings of the two training strategies. As seen in the table, with the same total compute budget of 64 hours and 128 GPUs, our low-to-high-res training strategy achieves significantly better performance, with a 2.6dB increase in PSNR. The key to enabling this is our fast low-res pretraining, which takes only 2.6 seconds per iteration with a larger batch size, allowing to train on more data within a shorter period of time and enabling much faster convergence. Benefiting from effective pretraining, our high-res finetuning can be trained with a smaller batch size that reduces the iteration time. In general, our 1st-stage training strategy significantly accelerates the LRM training, leading to high-quality NeRF reconstruction. This naturally improves the mesh reconstruction quality in the second stage.

Table 2: Effectiveness of our surface rendering fine-tuning stage. First row uses volume rendering; last two rows use surface rendering.

| | PSNR↑ | SSIM↑ | LPIPS↓ | CD↓ |
|---|---|---|---|---|
| MeshLRM (NeRF) | **28.13** | 0.923 | 0.093 | - |
| MeshLRM (NeRF)+MC | 25.48 | 0.903 | 0.097 | 3.26 |
| MeshLRM | 27.93 | **0.925** | **0.081** | **2.68** |

Table 3: Ablation study on the GSO dataset of losses used in the surface-rendering fine-tuning. CD is in units of $10^{-3}$.

| | PSNR↑ | SSIM↑ | LPIPS↓ | CD↓ |
|---|---|---|---|---|
| w/o Ray Opacity Loss | 18.46 | 0.836 | 0.218 | 51.98 |
| w/o Normal Loss | **27.93** | **0.925** | **0.080** | 2.87 |
| Ours (full model) | **27.93** | **0.925** | 0.081 | **2.68** |

**Effectiveness of surface fine-tuning (Stage 2).** We justify the effectiveness and importance of our 2nd-stage surface fine-tuning by comparing our final meshes with those directly extracted from the 1st-stage model using marching cubes on the GSO dataset. The quantitative results are shown in Tab. 2, and the qualitative results are presented in the Appendix F. Note that, while our 1st-stage NeRF reconstruction, i.e. MeshLRM (NeRF), can achieve high volume rendering quality, directly extracting meshes from it with Marching Cubes (MC), i.e. 'MeshLRM (NeRF)+MC', leads to a significant drop of rendering quality on all metrics. On the other hand, our final MeshLRM model, fine-tuned with DiffMC-based mesh rendering, achieves significantly better mesh rendering quality and geometry quality than the MeshLRM (NeRF)+MC baseline, notably increasing PSNR by 2.5dB and lowering CD by 0.58. This mesh rendering quality is even comparable — with a slight decrease

in PSNR but improvements in SSIM and LPIPS — to our 1st-stage volume rendering results of MeshLRM (NeRF).

**Surface fine-tuning losses.** We demonstrate the importance of the different losses applied during the mesh fine-tuning stage (details in Sec. 3.3) with an ablation study in Tab. 3 and Fig. 3. We observe a significant performance drop in the model without our proposed ray opacity loss, leading to severe floater artifacts as shown in Fig. 3. This is because, without supervision on empty space, the model may produce floaters to overfit to the input views rather than generating the correct geometry, as it does not receive a penalty for inaccuracies in empty space. Our normal loss helps to generate better geometry, i.e. a lower Chamfer distance. While it does not lead to significant quantitative improvements in rendering on the GSO dataset, we still observe qualitative improvements with this loss, especially when using generated images, as shown in Fig. 3 (right). Empirically, the normal loss leads to better robustness in handling inconsistent input views, a common challenge in text-to-3D or image-to-3D scenarios. Moreover, the geometric improvements from the normal loss are also crucial for real applications to use our meshes in 3D engines (like the one in Fig. 1), where accurate shapes are critical for high-fidelity physically-based rendering.

Table 4: Comparison between using big MLP and tiny MLP as triplane decoder. Both are trained for 60k iterations. $T_{\text{iter}}$ refers to the training time per iter. Results are volume rendering with Stage-1 256-res pretrained models.

| | PSNR↑ | SSIM↑ | LPIPS↓ | $T_{\text{iter}}$ |
|---|---|---|---|---|
| Triplane+Big MLP | 27.44 | **0.915** | 0.081 | 3.6s |
| Triplane+Tiny MLP | **27.47** | **0.915** | **0.080** | **2.7s** |

Figure 3: Ablation study on losses used in Stage-2 surface rendering fine-tuning.

w/o Ray Opacity Loss    w/o Normal Loss    w/ Normal Loss

**Tiny MLPs.** To justify our choice of using tiny MLPs for triplane decoding rather than the larger ones used in previous LRMs, we compare with a version that replaces the two tiny MLPs with a 10-layer (i.e., 9 hidden layers) 64-width shared MLP decoder (as used in (Hong et al., 2024; Li et al., 2024)). We show the quantitative results on the GSO dataset in Tab. 4. We find that the tiny MLP achieves similar performance to the large MLP while offering a notable training speed advantage (2.7s / step vs 3.6s / step, a 25% speedup). Moreover, we observe that the heavy shared MLP does not converge well during Stage 2 DiffMC training, unlike our proposed tiny separate MLPs.

**Removal of DINO.** We evaluated the benefits of DINO features and trained a MeshLRM model using the DINO encoder for the same number of steps as our proposed models. Despite having 85M (28%) more pre-trained parameters than the 300M in MeshLRM, the use of DINO features results in similar performance, with only a 0.1 dB difference in PSNR and a 0.001 difference in SSIM compared to our proposed model. We also found that using only Plücker Rays for positional encoding improves our model's zero-shot ability when processing higher resolutions (see the comparison in Appendix Fig. 8). We believe that this facilitates our high-res fine-tuning and leads to more efficient training.

### 4.3 COMPARISON AGAINST BASELINES

**Comparisons with feed-forward methods.**

We evaluate MeshLRM and other methods for 3D object reconstruction from four sparse views. As reported by Instant3D, previous cost-volume-based feed-forward methods (like SparseNeus(Long et al., 2022)) cannot handle this challenging sparse views case. We therefore focus on comparing with the LRM model in Instant3D (labeled as In3D-LRM) on the GSO dataset, showing quantitative results with detailed training settings and timings in Tab. 5. In the table, we show our results from both stages and compare them with both In3D-LRM and In3D-LRM + MC; for In3D-LRM + MC, Marching Cubes is directly applied to extract meshes from their NeRF reconstruction.

As shown in Tab. 5, our model, across both stages, achieves higher quality compared to In3D-LRM. As expected, In3D-LRM + MC leads to much worse rendering quality than In3D-LRM, due to the quality drop caused by the meshing process. Thanks to our effective 2nd-Stage DiffMC-based train-

Table 5: Comparison with feed-forward approaches on the GSO dataset. $T_{\text{infer}}$ is the wall-clock time from images to 3D representation (triplane for the first two rows and mesh for last two rows), and $T_{\text{train}}$ is the training budget; all reported using A100 GPUs. FPS is for rendering $512 \times 512$ resolution images from the corresponding representations. CD is in units of $10^{-3}$.

| | Render | FPS | $T_{\text{infer}}$ | Params | $T_{\text{train}}$ (GPU×day) | Quality (GSO) | | | |
| | | | | | | PSNR↑ | SSIM↑ | LPIPS↓ | CD↓ |
|---|---|---|---|---|---|---|---|---|---|
| In3D-LRM (Li et al., 2024) | 128 pts/ray | 0.5 | 0.07s | 500M | 128×7 | 26.54 | 0.893 | **0.064** | - |
| MeshLRM (NeRF) | 512 pts/ray | **2** | **0.06s** | 300M | **128×2.7** | **28.13** | **0.923** | 0.093 | - |
| In3D-LRM (Li et al., 2024)+MC | Mesh | >60 | 1s | 500M | 128×7 | 23.70 | 0.875 | 0.111 | 3.40 |
| MeshLRM | | >60 | **0.8s** | 300M | **128×3** | **27.93** | **0.925** | **0.081** | **2.68** |

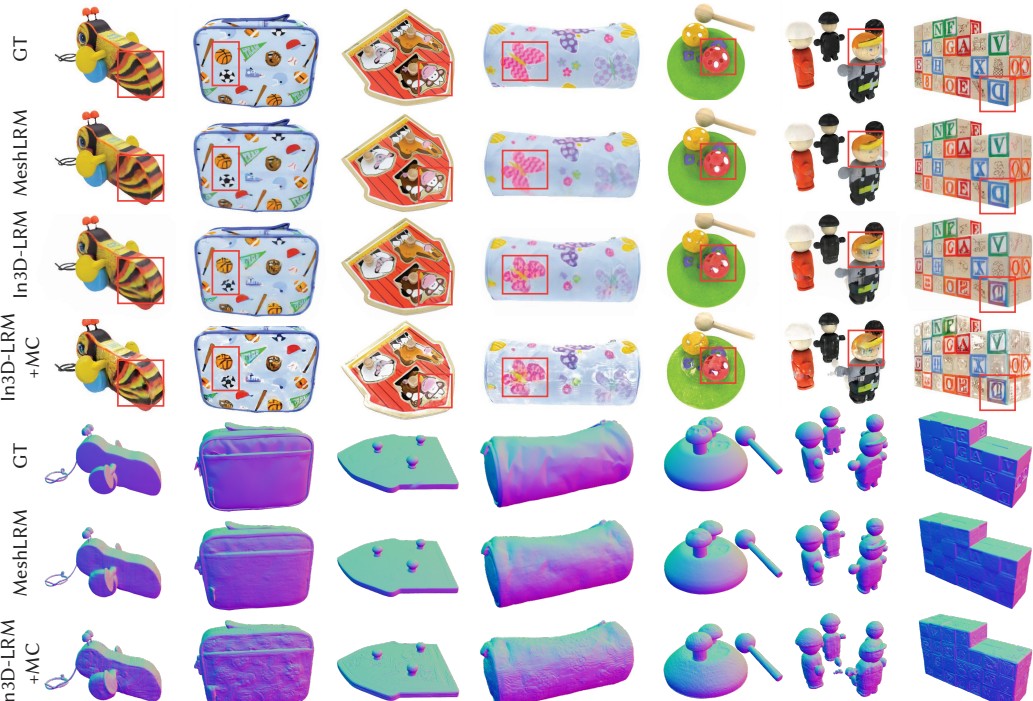

Figure 4: Qualitative comparison between MeshLRM and other feed-forward methods. 'In3D-LRM' is the Triplane-LRM in Instant3D (Li et al., 2024); 'MC' is Marching Cube. 'In3D-LRM' uses volume rendering and others use surface rendering.

ing, MeshLRM can reconstruct significantly better meshes in terms of both rendering quality (with 4.2dB PSNR and 0.05 SSIM improvements) and geometry quality (lowering the CD by 0.72). Our mesh rendering is even better than previous work volume rendering (In3D-LRM), notably increasing in PSNR by 1.4dB and SSIM by 0.03, while achieving substantially faster rendering speed. We show a qualitative comparison in Fig. 4 in which can see that our mesh renderings reproduce more texture and geometric details than the baselines. From Tab. 5, we can also see that our high reconstruction quality is achieved with smaller model size and substantially less compute, less than half of the total GPU×Day compute cost required for In3D-LRM. This is enabled by our simplified LRM architecture designs and efficient low-res-to-high-res 1st-stage training strategy, leading to significantly faster training speed and convergence. Besides, our model also leads to a faster inference speed, enabling high-quality mesh reconstruction within 1 second. Overall, our MeshLRM leads to state-of-the-art sparse-view mesh reconstruction quality with high parameter utilization efficiency, and faster training, inference, and rendering speeds. We note that our simplified MeshLRM (NeRF) also shows strong performance as a state-of-the-art LRM model for sparse-view NeRF reconstruction.

**Comparisons with per-scene optimization methods.**

We also compare our MeshLRM with recent per-scene optimization methods that are designed for sparse-view NeRF reconstruction, including FreeNeRF (Yang et al., 2023) and ZeroRF (Shi et al., 2023b). Our feed-forward method achieves better or comparable performance to methods that re-

Table 6: Comparison with single-image-to-3D methods on GSO and OmniObject3D datasets. CD is in units of $10^{-3}$.

| Method | GSO | | | | Omni3D | | | |
|---|---|---|---|---|---|---|---|---|
| | PSNR↑ | SSIM↑ | LPIPS↓ | CD↓ | PSNR↑ | SSIM↑ | LPIPS↓ | CD↓ |
| TripoSR (Tochilkin et al., 2024) | 19.85 | 0.753 | 0.265 | 27.48 | 17.68 | 0.745 | 0.277 | 28.69 |
| LGM (Tang et al., 2024) | 18.52 | 0.713 | 0.349 | 44.41 | 14.75 | 0.646 | 0.455 | 63.38 |
| InstantMesh (Xu et al., 2024) | 20.89 | 0.775 | 0.218 | 20.80 | 17.61 | 0.742 | 0.275 | 30.20 |
| MeshFormer (Liu et al., 2024c) | **21.47** | 0.793 | 0.201 | 16.30 | **18.14** | 0.755 | 0.266 | 25.91 |
| Ours | 21.31 | **0.794** | **0.192** | **15.74** | 18.10 | **0.759** | **0.257** | **25.51** |

quire minutes or hours to reconstruct a single scene, while taking only 0.8 seconds. Additional quantitative results and details are provided in Appendix B.

## 5 APPLICATIONS

Our approach achieves efficient and high-quality mesh reconstruction within one second, thus facilitating various 3D generation applications when combined with multi-view image techniques.

**Image-to-3D Generation** We follow the setup in the concurrent work MeshFormer (Liu et al., 2024c) to evaluate all methods (details in Appendix I), including TripoSR (Tochilkin et al., 2024), LGM (Tang et al., 2024), InstantMesh (Xu et al., 2024) and MeshFormer (Liu et al., 2024c). Specifically, for InstantMesh, MeshFormer, and our method, we first used Zero123++ (Shi et al., 2023a) to convert the input single-view image into multi-view images before performing 3D reconstruction. The other baselines follow their original configurations, taking a single-view image directly as input. The quantitative comparison results on GSO (Downs et al., 2022) and OmniObject3D (Wu et al., 2023) are shown in Tab. 6.

Our method demonstrates a significant advantage over TripoSR, LGM, and InstantMesh across all four metrics on both datasets. Additionally, it is comparable with MeshFormer, with slightly lower PSNR and better SSIM, LPIPS and CD. Note that MeshFormer is specifically designed for the single-image-to-3D task, it requires multi-view consistent normals as input during its reconstruction step. This makes it unsuitable for handling sparse-view reconstruction scenarios using captured images, such as setups with OpenIllumination, where only RGB images with camera poses are available. In contrast, our method can well handle this as shown in Tab. 7.

Additional visualization results are provided in Appendix Fig. 6, demonstrating that our generated meshes exhibit significantly higher quality, sharper textures, and more accurate geometric details.

**Text-to-3D Generation** We apply the multi-view diffusion models from Instant3D (Li et al., 2024) to generate 4-view images from text inputs, followed by our Mesh-LRM to achieve text-to-3D generation. Our results and comparison with the original Instant3D pipeline (In3D-LRM) (quantitatively compared in Tab. 5 in terms of reconstruction) are shown in the Appendix Fig. 7. As can be seen, our approach leads to better mesh quality and fewer rendering artifacts.

## 6 CONCLUSION

We present MeshLRM, a novel Large Reconstruction Model that directly outputs high-quality meshes. Our model leverages the Differentiable Marching Cubes (DiffMC) method and differentiable rasterization to fine-tune a pre-trained NeRF-based LRM, trained with volume rendering. As DiffMC requires backbone efficiency, we propose multiple improvements (tiny shared MLP and simplified image tokenization) to the LRM architecture, facilitating the training of both NeRF and mesh targeting models. We also find that a low-to-high-res training strategy significantly accelerates the training of the NeRF-based model. Compared with previous work, our method provides both quality and speed improvements and generates high-quality meshes. Finally, we show that our method can be directly applied to applications such as text-to-3D and image-to-3D generation. As meshes are the most widespread format for 3D assets in the industry, we believe our method takes a step towards better and faster integration of 3D asset creation networks in existing 3D workflows and manipulation tools.

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

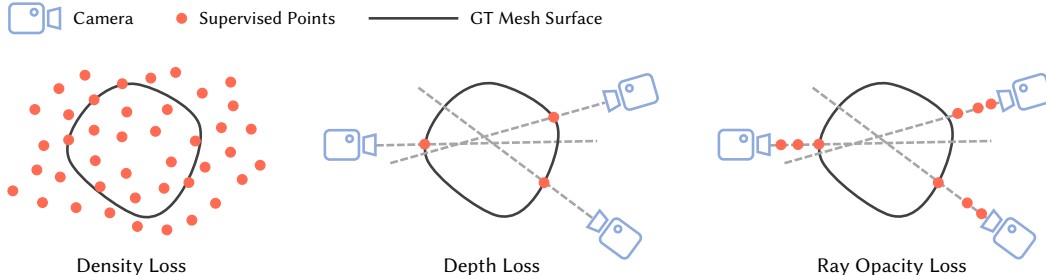

Figure 5: Illustration showing the difference on supervised points between our ray opacity loss with standard density loss and depth loss.

Table 7: Comparison with optimization approaches on Open-Illumination and NeRF-Synthetic datasets. Inference time is wall-clock time with a single A100 GPU. CD is in units of $10^{-3}$.

| | Inference Time | OpenIllumination PSNR↑ | SSIM↑ | LPIPS↓ | NeRF-Synthetic PSNR↑ | SSIM↑ | LPIPS↓ | CD↓ |
|---|---|---|---|---|---|---|---|---|
| FreeNeRF (Yang et al., 2023) | 3hrs | 12.21 | 0.797 | 0.358 | 18.81 | 0.808 | 0.188 | - |
| ZeroRF (Shi et al., 2023b) | 25min | 24.42 | 0.930 | 0.098 | **21.94** | **0.856** | 0.139 | 6.01 |
| MeshLRM | **0.8s** | **26.10** | **0.940** | **0.070** | 21.85 | 0.850 | **0.137** | **4.94** |

## APPENDIX

## A  DIFFERENT BETWEEN RAY OPACITY LOSS AND OTHER REGULARIZATIONS

The main difference between our ray opacity loss and previous approaches lies in the selection of supervision points. Standard density regularization uniformly supervises all points in space, including those on the mesh surface as well as in empty regions, which can lead to gradients that are less effective at guiding the model toward a clean surface. On the other hand, standard depth loss only supervises points on the mesh surface; once floaters appear during training, its gradients can move these floaters but cannot easily eliminate them. In contrast to these standard losses, our opacity loss supervises the entire space between the surface and the camera, effectively preventing floaters and stabilizing the training process.

## B  COMPARISON WITH PER-SCENE OPTIMIZATION METHODS

We compare our feed-forward sparse-view reconstruction method with FreeNeRF (Yang et al., 2023) and ZeroRF (Shi et al., 2023b) on NeRF-Synthetic (Mildenhall et al., 2020) and OpenIllumination datasets (Liu et al., 2024a).

Due to their use of per-scene optimization, these methods are much slower, requiring tens of minutes and up to several hours to reconstruct a single scene. As a result, it is not practical to evaluate them on the GSO dataset, comprising more than 1000 objects. Therefore, we conduct this experiment on the NeRF-Synthetic and OpenIllumination datasets, following the settings used in ZeroRF (Shi et al., 2023b) with four input views. Tab. 7 shows the quantitative novel view synthesis results, comparing our mesh rendering quality with their volume rendering quality. Since ZeroRF uses a post-processing step to improve their mesh reconstruction from Marching Cubes, we compare our mesh geometry with theirs using the CD metric on the NeRF-Synthetic dataset (where GT meshes are accessible).

As shown in Tab. 7, our NeRF-Synthetic results significantly outperform FreeNeRF; we also achieve comparable rendering quality and higher geometry quality to ZeroRF in a fraction of the execution time. On the challenging OpenIllumination real dataset, our approach outperforms both FreeNeRF and ZeroRF by a large margin, consistently for all three metrics. The OpenIllumination dataset, consisting of real captured images, demonstrates our model's capability to generalize to real captures for practical 3D reconstruction applications, despite being trained only on rendered images. In addition to our superior quality, our feed-forward mesh reconstruction approach is significantly faster than

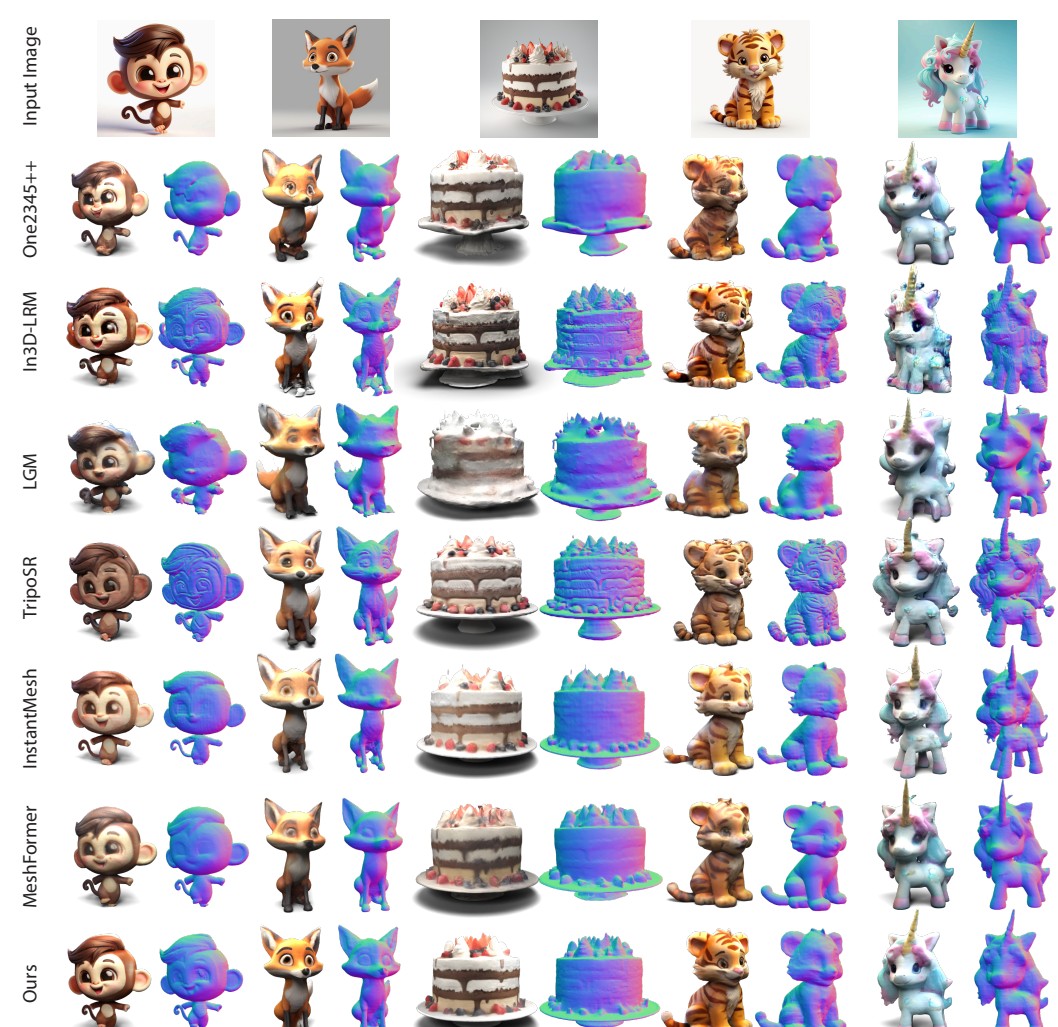

Figure 6: Image-to-3D comparison with other baseline methods. We utilize Zero123++ Shi et al. (2023a) to generate six-view images from a single input image. Note that our model is trained on 4 views and can zero-shot generalize to 6 views and achieve better results than other image-to-3D methods.

these optimization-based NeRF methods, in terms of both reconstruction ($1000\times$ to $10000\times$ speed up) and rendering.

## C APPLICATION VISUALIZATIONS

Our feed-forward reconstruction model can be used for 3D generation from text or single image prompts. We apply the multi-view diffusion models from Instant3D (Li et al., 2024) for text-to-multi-view generation and Zero123++ (Shi et al., 2023a) for image-to-multi-view generation. A visual comparison of our method with other 3D generation methods is shown in Fig. 7 and Fig. 6.

## D INFERENCE ON VARIOUS NUMBERS OF VIEWS

Although our model is trained with only 4 input views, it is capable of handling varying numbers of views during inference. We evaluate MeshLRM on 2-8 input views using the GSO dataset, as presented in Tab. 8. As the number of input views increases, the model demonstrates consistent

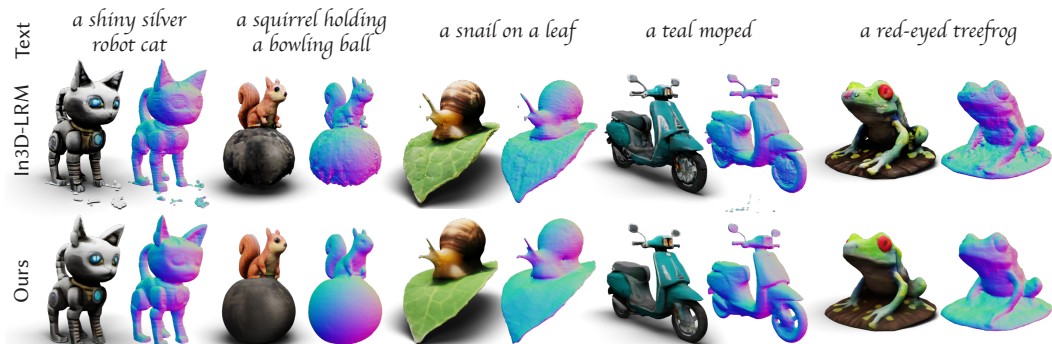

Figure 7: Text-to-3D results by applying Instant3D's (Li et al., 2024) diffusion model to generate 4-view images from text input. Our method can generate significantly more accurate and smoother geometry, along with sharp textures.

performance improvements, while still producing strong results even with just two input views, highlighting the robustness and flexibility of our approach.

## E    EFFECTIVENESS OF REMOVING DINO ENCODER.

We evaluate our model alongside In3D-LRM Li et al. (2024) (both trained on $512 \times 512$ resolution images) on $1024 \times 1024$ images, as shown in Fig 8. We observe significant geometry distortion in the In3D-LRM results, while our model still produces reasonable outputs. This demonstrates that our choice of positional encoding, which eliminates the DINO encoder and relies solely on Plücker Rays, greatly improves the model's robustness to changes in image resolution.

Table 8: Our MeshLRM trained on four input views generalizes to other view numbers. (Results on GSO)

| # views | 2 | 3 | 4 | 6 | 8 |
|---|---|---|---|---|---|
| PSNR↑ | 24.85 | 26.91 | 27.93 | 29.09 | 29.35 |
| SSIM↑ | 0.897 | 0.916 | 0.925 | 0.935 | 0.938 |
| LPIPS↓ | 0.099 | 0.087 | 0.081 | 0.071 | 0.069 |

Figure 8: Results with 1024x1024 (untrained resolution) input. Both models are trained at 512x512. In3D-LRM has obvious artifacts which our method doesn't suffer from.

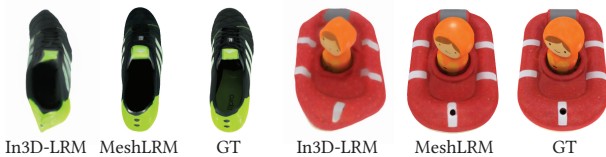

In3D-LRM    MeshLRM    GT       In3D-LRM    MeshLRM    GT

## F    EFFECTIVENESS OF SURFACE FINE-TUNING (STAGE 2).

We discuss the effectiveness of our second-stage fine-tuning in Sec. 4.2 with quantitative results shown in Tab. 2 in the main paper. We now show examples of qualitative results in Fig. 9, comparing our final meshes (MeshLRM) with the meshes (MeshLRM (NeRF) + MC) generated by directly applying Marching Cubes on the 1st-stage model's NeRF results. As shown in the figure, the MeshLRM (NeRF) + MC baseline leads to severe artifacts with non-smooth surfaces and even holes, while our final model with the surface rendering fine-tuning can effectively address these issues and produce high-quality meshes and realistic renderings. These qualitative results demonstrate the large visual improvements achieved by our final model for mesh reconstruction, reflecting the big quantitative improvements shown in the paper.

## G    ABO AND OPENILLUMINATION DATASETS

We compare our proposed method with In3D-LRM (Li et al., 2024) on 1000 randomly sampled examples in the ABO (Collins et al., 2022) dataset with quantitative results shown in Tab. 9. In particular, ABO is a highly challenging synthetic dataset that contains many objects with complex glossy materials. This is a big challenge for our model, as well as In3D-LRM's model, since both assume Lambertian appearance. In this case, the NeRF models are often better than the mesh models, since NeRF rendering can possibly fake glossy effects by using colors from multiple points

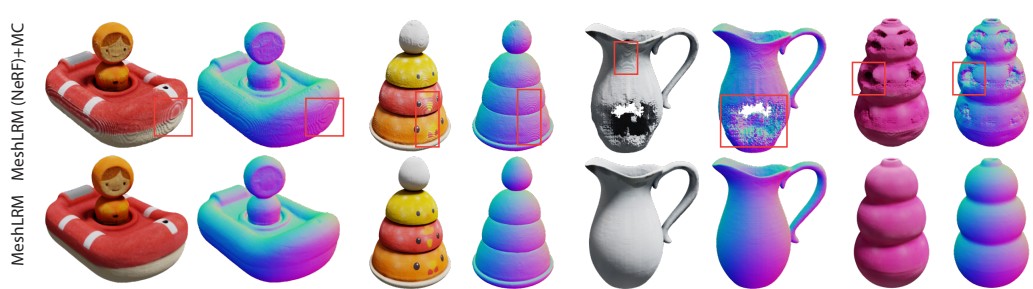

Figure 9: The reconstructed mesh quality drops significantly when applying marching cubes to the volume rendering trained model ('MeshLRM (NeRF) + MC') without our mesh refinement fine-tuning ('MeshLRM').

Table 9: Comparison with feed-forward approaches on ABO (Collins et al., 2022) and Open-Illumination (Liu et al., 2024a) datasets.

| | ABO | | | OpenIllumination | | |
|---|---|---|---|---|---|---|
| | PSNR↑ | SSIM↑ | LPIPS↓ | PSNR↑ | SSIM↑ | LPIPS↓ |
| In3D-LRM (Li et al., 2024) | 27.50 | 0.896 | 0.093 | 8.96 | 0.568 | 0.682 |
| MeshLRM (NeRF) | **28.31** | **0.906** | **0.108** | **20.53** | **0.772** | **0.290** |
| In3D-LRM (Li et al., 2024)+MC | 22.11 | 0.850 | 0.144 | 8.92 | 0.507 | 0.691 |
| MeshLRM | **26.09** | **0.898** | **0.102** | **20.51** | **0.786** | **0.218** |

along each pixel ray while mesh rendering cannot. Nonetheless, as shown in the table, both our NeRF and final mesh model can outperform In3D-LRM's NeRF and mesh variants respectively. Especially, thanks to our effective MC-based finetuning, our mesh rendering quality surpasses that of In3D-LRM + MC by a large margin (e.g. 3.92dB in PSNR and 0.48 in SSIM).

We also compare with In3D-LRM on the full OpenIllumination (Liu et al., 2024a) dataset in Tab. 9. This experiment setting is different from the setting used in Tab. 6 in the main paper, where we follow ZeroRF (Shi et al., 2023b) to test only 8 objects with combined masks in OpenIllumination; here, we evaluate all models on the full dataset with 100 objects with object masks and take square crops that tightly bound the objects from the rendered/GT images to compute rendering metrics, avoiding large background regions. As shown in the table, we observe that In3D-LRM cannot generalize to this challenging real dataset, leading to very low PSNRs, despite being trained on the same training data as ours. In contrast, our model still works well on this out-of-domain dataset with high rendering quality.

## H  ADDITIONAL IMPLEMENTATION DETAILS

The transformer size follows the transformer-large config (Devlin et al., 2019). It has 24 layers and a model width of 1024. Each attention layer has a 16 attention head and each head has a dim of 64. The intermediate dimension of the MLP layer is 4096. We use GeLU activation inside the MLP and use Pre-LN architecture. The layer normalization is also applied after the input image tokenization, and before the triplane's unpachifying operator.

For the training of both stages, we use AdamW with $\beta_2 = 0.95$. Other Adam-specific parameters are set as default and we empirically found that the model is not sensitive to them. A weight decay of 0.05 is applied to all parameters excluding the bias terms and layer normalization layers. The learning rate for stage 1 is $4e - 4$. We also use cosine learning rate decay and a linear warm-up in the first 2K steps. For stage 2, we use a learning rate of $1e - 5$, combined with cosine learning rate decay and a linear warm-up during the first 500 steps. In total, there are 10k fine-tuning steps. The resolution of DiffMC is 256 within a $[-1, 1]^3$ bounding box, which is consistent with the triplane resolution.

For the CD metric, we sample points only on the visible surface from the testing views. Since the ground-truth meshes from GSO Downs et al. (2022) and OmniObject3D Wu et al. (2023) are obtained through scanning, their interior structures cannot be considered reliable. For example, we found that the interiors of shoes in GSO are incorrectly represented, with a face filling the interior, which contradicts common sense. To address this, we cast rays from all testing views and sample 100,000 points at the ray-surface intersections for each object, following a strategy similar to that used in Tang et al. (2022).

## I DETAILS FOR IMAGE-TO-3D TESTING SETUP

For the GSO dataset, the single-view input is the first thumbnail image, while for the OmniObject3D dataset, the input is a rendered image with a randomly selected pose. We select 24 camera poses evenly distributed around the object to capture a full 360-degree view and use BlenderProc to render images at a resolution of 320×320. The object is normalized to fit within a $[-1, 1]^3$ bounding box. The camera positions are defined by azimuth angles of [0, 45, 90, 135, 180, 225, 270, 315] degrees and elevation angles of [60, 90, 120] degrees.

## J LIMITATIONS

As our model employs surface rendering and assumes Lambertian appearance without performing inverse rendering, it is not sufficiently robust when the input images contain complex materials, such as the metal objects shown in Fig. 10. The generated mesh may bake the shadow and use white color to fit the specular areas. We believe that incorporating inverse

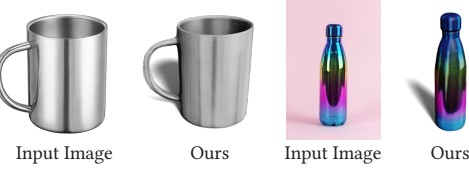

Input Image    Ours    Input Image    Ours

Figure 10: Failure cases of MeshLRM.

rendering into the current pipeline could address this issue. However, this requires sufficient training data with accurate material annotations, which we leave for future exploration. On the other hand, while handling highly sparse input views, our model requires input camera poses. Although poses can be readily obtained from text- or image-to-multi-view models (Li et al., 2024; Shi et al., 2023a) for 3D generation tasks, calibrating sparse-view poses for real captures is a challenge. In the future, it will be an interesting direction to explore combining our approach with recent pose estimation techniques (Wang et al., 2023a;b) for joint mesh reconstruction and camera calibration.

