# OpenReview forum: "MeshLRM: Large Reconstruction Model for High-Quality Meshes"
_ICLR.cc/2025/Conference — Submitted to ICLR 2025_

### Official Review · Reviewer_1jbs · 2024-10-21

**Soundness:** 2
**Presentation:** 3
**Contribution:** 2
**Rating:** 5
**Confidence:** 5

**Summary:**

This manuscript introduces a two-stage feed-forward large reconstruction model that utilizes sparse views as input. In the first stage, NeRF-like volume rendering is employed for supervision, with progressive training regarding resolution. In the second stage, Differentiable Marching Cubes (DiffMC) is leveraged to convert the density field into a mesh, which supports fast rendering and geometry supervision. With the integration of a ray-opacity loss, MeshLRM is capable of generating high-quality meshes. Furthermore, by simplifying the encoder, MeshLRM achieves faster inference speeds.

**Strengths:**

1. Utilizing DiffMC in LRM is novel. Indeed, meshes offer superior 3D representation.
2. The reconstructed mesh appears to be of high quality, featuring smooth geometry.
3. The proposed ray-opacity loss is technically sound and well-founded.

**Weaknesses:**

1. The definition of post-optimization in Line 56 is unclear. In traditional multi-view reconstruction tasks with SDF-based volume rendering, meshes can be directly extracted by Marching Cubes (MC) by identifying the zero-level set without any post-optimization.
2. A discussion is lacking on why converting NeRFs into meshes leads to a significant drop in rendering quality and geometric accuracy, as mentioned in Line 66.
3. The definitions of the post-processing steps, specifically Marching Cubes in NeRF and Differentiable Marching Cubes in MeshLRM, should be illustrated more clearly. If both processes are considered post-processing, then the disadvantages of NeRF should not include the need for additional post-optimization.
4. There are some grammatical issues. For example, a comma is missing in Line 76.
5. I recommend adding an illustration in Line 75 to depict why NeRF-based LRM pretraining is important.
6. Considering there are two-stage optimizations, the claim of end-to-end optimization in Line 83 is not precise enough.
7. The assertion in Line 205 is not convincing enough. A comparison with In3D-LRM does not substantiate the claim that DINO cannot handle high-resolution images. There should be an ablation study in MeshLRM to adequately discuss why DINO is not necessary. Since you claim a contradiction in this paragraph, it is crucial to address this.
8. There is no quantitative comparison between NeRF-based LRM and Mesh-based LRM. Additionally, it appears that NeRF-based LRM achieves better PSNR.
9. There is a lack of quantitative comparison in the progressive training regarding resolution.
10. There are too few comparison methods. Only LGM and In3D are compared.

**Questions:**

1. Why are density fields considered more compatible with the Marching Cubes step in Line 73? In traditional multi-view reconstruction tasks, Signed Distance Functions (SDFs) have proven to be more effective than density fields for mesh reconstruction, as surfaces are clearly defined by the zero-level set. What makes density fields preferable in this context?

2. I  do not fully understand the challenges associated with the second-stage mesh-based LRM optimization. When using Flexicube in Instantmesh, there do not appear to be issues related to mesh floater. Could you elaborate on the advantages of choosing Differentiable Marching Cubes (DiffMC) over Flexicube?

---

> ### Author Response · Authors · 2024-11-17
>
> Abbreviations in the answers: W: Weaknesses; Q: Questions
>
> **Clarification about post-optimization (W1, W3):**
> Post-optimization in Line 56 refers to techniques applied after training a neural radiance field, requiring a separate mesh-related optimization stage, such as fine-tuning (nerf2mesh [1], NeuManifold [2]) or distillation (Nerfmeshing [3]), to enhance mesh reconstruction quality in the per-scene optimization case. It does not refer to directly applying marching cubes to the density field.
>
> **Why converting NeRF-based method leads to performance drop (W2):**
> Since volume rendering does not require a strict surface definition, applying marching cubes to extract a mesh surface can easily lead to inaccuracies in mesh quality, such as geometric noise or artifacts (e.g., ringing effects). This is why previous works [1][2][3] try to address this in the per-scene optimization task.
>
> **Why NeRF-based pretraining is necessary (W5):**
> We found that single-stage training (i.e., without NeRF initialization) fails to produce reasonable results. Directly training using a mesh representation often gets stuck in local minima and fail to converge, whereas neural radiance fields provide a better representation for handling topological changes in a smooth, continuous manner. This issue has also been widely noted and discussed in several per-scene optimization studies, such as Magic3D [1], NeuManifold [2], and Nerf2Mesh [3].
>
> **Statement of end-to-end optimization (W6):**
> Although our training process is divided into two stages, the network architecture remains consistent. And our second-stage mesh training optimizes all components jointly, directly regressing the final mesh output from the input images, which is completely end-to-end.
>
> **Why DINO is not necessary (W7):**
> In addition to comparing with In3D, we also evaluated the benefits of DINO features and trained a MeshLRM model using the DINO encoder for the same number of steps as our proposed models, which leads to negligible quality differences. More details can be seen in paper line 405.
>
> **Quantitative comparison between NeRF-based LRM and Mesh-based LRM (W8):**
> We provide a quantitative comparison in Table 5 between MeshLRM (NeRF) and MeshLRM. Our MeshLRM achieves comparable results to MeshLRM (NeRF), with even better performance on SSIM and LPIPS metrics. This is particularly challenging because, in per-scene optimization studies, mesh-based methods typically exhibit lower quality compared to volumetric-based methods, as shown in nvdiffrec [9] and nerf2mesh [7].
>
> **Quantitative comparison in the progressive training regarding resolution (W9):**
> We provide a quantitative comparison in Table 1, contrasting training from scratch on high-resolution data with our low-to-high-resolution training strategy.

---

> ### Author Response · Authors · 2024-11-17
>
> **Comparison with newer baseline (W10):**
> We have added a new table to quantitatively compare our method against recent baselines. Given that input settings differ among the baselines, we follow the setup in the concurrent work MeshFormer [4] to evaluate all methods under a unified single-view-to-3D setting. Specifically, for InstantMesh, MeshFormer, and our method, we first used Zero123++ [5] to convert the input single-view image into multi-view images before performing 3D reconstruction. The other baselines follow their original configurations, taking a single-view image directly as input.
>
> The results, where all methods generate a mesh as the final output, are presented below.
> | Method           |        | GSO                |        |        | Omni3D             |        |
> |------------------|--------|--------------------|--------|--------|--------------------|--------|
> |                  | PSNR   | SSIM               | LPIPS  | PSNR   | SSIM               | LPIPS  |
> | TripoSR [1]      | 19.85  | 0.753              | 0.265  | 17.68  | 0.745              | 0.277  |
> | LGM [2]          | 18.52  | 0.713              | 0.349  | 14.75  | 0.646              | 0.455  |
> | InstantMesh [3]  | 20.89  | 0.775              | 0.218  | 17.61  | 0.742              | 0.275  |
> | MeshFormer [4]   | 21.47  | 0.793              | 0.201  | 18.14  | 0.755              | 0.266  |
> | Ours             | 21.31  | 0.794              | 0.192  | 18.10  | 0.759              | 0.257  |
>
> Our method demonstrates a significant advantage over TripoSR, LGM, and InstantMesh across all three metrics on both datasets. Additionally, it is comparable with MeshFormer, with slightly lower PSNR and better SSIM and LPIPS. Note that MeshFormer is specifically designed for the single-image-to-3D task, it requires multi-view consistent normals as input during its reconstruction step. This makes it unsuitable for handling sparse-view reconstruction scenarios using captured images, such as setups with OpenIllumination, where only RGB images with camera poses are available. In contrast, our method can well handle this as shown in Table 6.
>
> We will also expand the discussion on recent works in the related work section and include the new results in the revised version.
>
> **What makes density fields preferable in this context? (Q1)**
> Based on the study on per-scene optimization papers, SDF-based methods, suggested by the reviewer, tend to produce over-smoothed geometry and struggle to capture thin structures, as noted by Nerf2Mesh [7], and they also achieve worse rendering quality compared to density-based methods. In contrast, the strategy of pretraining with density fields followed by fine-tuning with rasterization has been demonstrated to be effective in previous works, such as Magic3D [6], NeRF2Mesh [7], and NeuManifold [8]. Therefore, we have chosen to adopt this approach. We successfully make our mesh rendering quality match the volume rendering quality, which is rare in per-scene optimization methods.
>
> **The reason for choosing DiffMC (Q2):**
> Our pipeline is compatible with DMTet and Flexicubes, not limited to DiffMC. However, we chose DiffMC due to its superior processing speed and minimal GPU VRAM consumption (as detailed in the comparison table on its GitHub repository: https://github.com/SarahWeiii/diso?tab=readme-ov-file#speed-comparison). This efficiency is particularly beneficial for large-scale training, where both speed and GPU optimization are crucial.
>
> Compared to the concurrent work InstantMesh, there are numerous differences between our pipeline and theirs. Factors such as model architecture, model size, training precision, and other design choices can significantly impact the training difficulty. However, we have carefully addressed the challenges in training our model, resulting in a method that achieves better quality than InstantMesh [3].
>
>
>
> [1] Tochilkin, Dmitry, et al. "Triposr: Fast 3d object reconstruction from a single image." arXiv (2024).
>
> [2] Tang, Jiaxiang, et al. "Lgm: Large multi-view gaussian model for high-resolution 3d content creation." ECCV, 2025.
>
> [3] Xu, Jiale, et al. "Instantmesh: Efficient 3d mesh generation from a single image with sparse-view large reconstruction models." arXiv (2024).
>
> [4] Liu, Minghua, et al. "Meshformer: High-quality mesh generation with 3d-guided reconstruction model." NeurIPS (2024).
>
> [5] Shi, Ruoxi, et al. "Zero123++: a single image to consistent multi-view diffusion base model." arXiv (2023).
>
> [6] Lin, Chen-Hsuan, et al. "Magic3d: High-resolution text-to-3d content creation." CVPR. 2023.
>
> [7] Tang, Jiaxiang, et al. "Delicate textured mesh recovery from nerf via adaptive surface refinement." ICCV. 2023.
>
> [8] Wei, Xinyue, et al. "Neumanifold: Neural watertight manifold reconstruction with efficient and high-quality rendering support." arXiv (2023).
>
> [9] Munkberg, Jacob, et al. "Extracting triangular 3d models, materials, and lighting from images." CVPR. 2022.

---

> > ### Comment · Reviewer_1jbs · 2024-11-20
> >
> > Since MeshLRM is essentially a reconstruction model, I believe that comparisons based solely on 2D metrics are insufficient. Including 3D metrics such as Chamfer distance and F-score would provide a more comprehensive evaluation.
> >
> > Furthermore, given the stated limitation that ``it is not sufficiently robust when the input images contain complex materials,'' I think it would be helpful to include a visualization of a failure case to illustrate this limitation.

---

> > > ### Author Response · Authors · 2024-11-22
> > >
> > > We compute the mesh metric for this table using the same setup as in other experiments presented in the paper. The Chamfer distance is calculated using points sampled from both the ground-truth and reconstructed meshes within a $[-1, 1]^3$ bounding box, measured in units of $10^{-3}$.
> > >
> > > | Method          |       | GSO   |       |       |       | Omni3D |       |       |
> > > |-----------------|-------|-------|-------|-------|-------|--------|-------|-------|
> > > |                 | PSNR  | SSIM  | LPIPS | CD    | PSNR  | SSIM   | LPIPS | CD    |
> > > | TripoSR [1]     | 19.85 | 0.753 | 0.265 | 27.48 | 17.68 | 0.745  | 0.277 | 28.69 |
> > > | LGM [2]         | 18.52 | 0.713 | 0.349 | 44.41 | 14.75 | 0.646  | 0.455 | 63.38 |
> > > | InstantMesh [3] | 20.89 | 0.775 | 0.218 | 20.80 | 17.61 | 0.742  | 0.275 | 30.20 |
> > > | MeshFormer [4]  | 21.47 | 0.793 | 0.201 | 16.30 | 18.14 | 0.755  | 0.266 | 25.91 |
> > > | Ours            | 21.31 | 0.794 | 0.192 | 15.74 | 18.10 | 0.759  | 0.257 | 25.51 |
> > >
> > > As for the failure cases, we are happy to add a figure for visualization in the revised paper.

---

> > > > ### Comment · Reviewer_1jbs · 2024-11-25
> > > >
> > > > Thank you for providing the visualization of the failure cases.
> > > >
> > > > I have a minor concern. While trying the online Gradio demo, I noticed that the geometry of the cup shown in the failure case appears nearly perfect. However, this seems inconsistent with the results presented in the revised manuscript, where the cup exhibits noticeable uneven surfaces.
> > > >
> > > > Could you clarify if there are any differences between the unofficial online demo and the original MeshLRM? Could it possibly be an updated version of MeshLRM?

---

> > > > > ### Author Response · Authors · 2024-11-25
> > > > >
> > > > > We tested the unofficial online Gradio demo using the cup model and observed that the rendering effect produced by the Gradio mesh renderer differs from that of Blender, which we used to render images for the paper. Multiple factors may contribute to this difference, such as lighting conditions, shading algorithms, and material properties used in the rendering process. Although the mesh appears smooth in the Gradio visualizer, small uneven artifacts are still visible when the downloaded mesh is viewed in our Blender setup. Additionally, we noticed that the lighting conditions we initially chose in Blender magnified the artifacts, so we updated the figure with improved lighting conditions.

---

> > > > > > ### Comment · Reviewer_1jbs · 2024-11-27
> > > > > >
> > > > > > I still have concerns about the performance difference between the Gradio demo and MeshLRM. In the first version, there is a noticeable concave on the surfaces, and I don’t think that lighting can significantly alter the geometry.
> > > > > >
> > > > > > I believe surface normals provide a more accurate representation of the geometry. Therefore, I recommend using normal visualization for failure cases, rather than relying on rendered RGB images.

---

> > > > > > > ### Author Response · Authors · 2024-11-27
> > > > > > >
> > > > > > > The lighting does not change the geometry but rather affects its appearance (shading and shadowing). See the [video](https://meshlrm.github.io/videos/lighting.mp4) demonstrating how lighting influences rendering. In the video, we change the lighting condition from the current one to the previous one and then back again, during which the artifacts gradually appear and then disappear.
> > > > > > >
> > > > > > > We can include the surface normal in the final paper, as suggested by the reviewer ([figure](https://meshlrm.github.io/videos/failure.pdf)). From the surface normal, it can be seen that the artifacts are at similar levels to the image renderings. This is why we stated that the previous lighting condition *magnifies* the artifacts.
> > > > > > >
> > > > > > > (Click the "video" and "figure" links to view the results in anonymous links.)

---

> ### Comment · Reviewer_1jbs · 2024-12-01
>
> After reading through the responses and review suggestions from other reviewers, I noticed that all reviewers raised concerns about the novelty of this work. Reviewer ftqh specifically pointed out the lack of public availability.
>
> While the authors argue that the simplification of the network architecture and the regularization loss are carefully designed, I believe that without DiffMC and using Flexicube as InstantMesh, we wouldn’t need the regularization loss at all.
>
> Furthermore, although Reviewer 6QBv mentions that "While combining LRM with diffMC is an effective and practical engineering approach, the concept appears somewhat ad hoc. However, this does not constitute a significant reason for rejection, as the paper still meets the typical standards of previously accepted work.'', I do not fully agree with this point. I do not think that LRMs or 3D generation should be held to a lower standard for acceptance.
>
> Overall, considering all of these factors, we have decided to revise my rating to a weak reject.

---

> ### Author Response · Authors · 2024-12-02
>
> We are glad to have addressed the earlier concerns raised by the reviewer during the discussion. However, we are very surprised that new concerns, which were not previously mentioned or emphasized, have now been introduced and considered significant enough to lower the rating. We now provide quick responses to address these points.
>
> **Public Availability:** The reviewer has already discovered and successfully tested a publicly available unofficial demo of our method, which shows similar quality. This clearly demonstrates the reproducibility and availability of our approach. We promise to release a public demo implemented by the authors to ensure availability.
>
> **Novelty and Contribution:** While we acknowledge varying perspectives on assessing a paper’s contribution, we believe the most critical measure—and the fundamental goal of research—is advancing the state of the art in the field. Our work achieves this by outperforming all baseline methods, including very recent approaches such as LGM (ECCV 2024) and MeshFormer (NeurIPS 2024), both of which were recognized as state of the arts and published within months. This highlights the significance of our contribution. We believe that any work that genuinely advances the state of the art with carefully justified experiments deserves to be seen by the community. Furthermore, we strongly disagree with the characterization of our work as "ad hoc" or "trivial engineering." Every design choice in our framework is carefully evaluated and supported by experimental evidence. If any aspect remains unclear, we are happy to provide additional clarifications.
>
> **Regarding InstantMesh:** InstantMesh is a concurrent work with high-level similarities to ours. In principle, such concurrent works should not influence the evaluation of our paper. Nevertheless, we have already provided extensive discussions and comparisons with InstantMesh, demonstrating that our method achieves superior quality. Specifically, we emphasize that our opacity loss is a novel design in our pipeline, as shown in newly added Figure 5, with experimental evidence in Figure 3 confirming its necessity and effectiveness. Designing a new model that integrates elements from both our work and InstantMesh to achieve high quality while removing the opacity loss goes beyond the scope of this submission.
>
> We believe unbiased reviewing should focus on the demonstrated quality, experiments, and contributions of the submission itself. If there are specific points that remain critical to alter the rating, we kindly ask for clarification so we can address them in detail.

---

### Official Review · Reviewer_ftqh · 2024-11-02

**Soundness:** 3
**Presentation:** 4
**Contribution:** 3
**Rating:** 6
**Confidence:** 5

**Summary:**

This paper presents MeshLRM, a large reconstruction model that enables the generation of high-quality 3D meshes from posed sparse-view images. It incorporates differentiable mesh extraction and rendering within the LRM framework, simplifying the architecture and improving training strategies for faster and better quality reconstruction. The model introduces a ray opacity loss for stable DiffMC-based training and achieves state-of-the-art performance in sparse-view mesh reconstruction, supporting applications like text-to-3D and single-image-to-3D generation.

**Strengths:**

- The incorporation of differentiable mesh extraction and rendering within the LRM framework is reasonable, enabling end-to-end mesh reconstruction without post-optimization steps.
- The introduction of the ray opacity loss is a novel contribution that addresses the challenge of stabilizing training for mesh reconstruction.
- The paper is well-written and easy to follow. The mesh visualization in the teaser is impressive.

**Weaknesses:**

- Limited evaluation on real-World data.
The paper primarily evaluates MeshLRM on synthetic datasets. While the performance on these datasets is impressive, the model's ability to generalize to real-world images with complex lighting and materials is less explored. Incorporating more experiments on real-world datasets such as OmniObject3d [1] could provide a clearer picture of the model's practical applicability.

- Incomplete discussion and comparison with newer baselines.
The paper compares with Instant3D and LGM, the former is not open-sourced, while the latter is based on Gaussian Splatting and leverages a ConvNet instead of transformer. I think there are some newer methods should be discussed and compared in the paper, e.g., InstantMesh [2], GeoLRM [3] and MeshFormer [4]. Especially, I believe InstantMesh is a pefect baseline for comparison since it is open-sourced and also adopts the "LRM+differentiable mesh extraction" paradigm.

- Concerns on technical novelty and training stability.
Pioneer works like Magic3D [5] and InstantMesh have explored the pipeline of two-stage 3D generation by combining the NeRF and mesh representations, so I think the usage of mesh representation is reasonable but not novel.
Besides, I have another concern on the training stability of mesh representation. According to the paper, the training process of MeshLRM is splitted into two stages, using NeRF and mesh respectively, and "Once trained with volume rendering, our model already achieves high-fidelity sparse-view NeRF reconstruction, which can be rendered with ray marching to create realistic images" (Section 3.3, line 272). Why is stage 1 essential if the proposed ray opacity loss can indeed stablize the training process on the mesh representation? What if we skip stage 1 and directly train the model on the mesh representation from scratch? I think the authors should add some discussions on this issue in the paper.

[1] Wu, Tong, et al. "Omniobject3d: Large-vocabulary 3d object dataset for realistic perception, reconstruction and generation." Proceedings of the IEEE/CVF Conference on Computer Vision and Pattern Recognition. 2023.
[2] Xu, Jiale, et al. "Instantmesh: Efficient 3d mesh generation from a single image with sparse-view large reconstruction models." arXiv preprint arXiv:2404.07191 (2024).
[3] Zhang, Chubin, et al. "Geolrm: Geometry-aware large reconstruction model for high-quality 3d gaussian generation." arXiv preprint arXiv:2406.15333 (2024).
[4] Liu, Minghua, et al. "Meshformer: High-quality mesh generation with 3d-guided reconstruction model." arXiv preprint arXiv:2408.10198 (2024).
[5] Lin, Chen-Hsuan, et al. "Magic3d: High-resolution text-to-3d content creation." Proceedings of the IEEE/CVF Conference on Computer Vision and Pattern Recognition. 2023.

**Questions:**

My questions have been listed in the weakness part, the author's response on them will serve as the main basis for my final rating. Besides, the paper does not mention the future availability of code or pre-trained models, which is crucial for reproducibility and further research by the community. Providing such resources would greatly benefit the field.

---

> ### Author Response · Authors · 2024-11-17
>
> Abbreviations in the answers: W: Weaknesses; Q: Questions
>
> **Comparison with newer baseline and on real-world dataset (W1, W2):**
> We have added a new table to quantitatively compare our method against recent baselines both on GSO and OmniObject3D datasets. Given that input settings differ among the baselines, we follow the setup in the concurrent work MeshFormer [4] to evaluate all methods under a unified single-view-to-3D setting. Specifically, for InstantMesh, MeshFormer, and our method, we first used Zero123++ [5] to convert the input single-view image into multi-view images before performing 3D reconstruction. The other baselines follow their original configurations, taking a single-view image directly as input.
>
> The results, where all methods generate a mesh as the final output, are presented below.
>
> | Method           |        | GSO                |        |        | Omni3D             |        |
> |------------------|--------|--------------------|--------|--------|--------------------|--------|
> |                  | PSNR   | SSIM               | LPIPS  | PSNR   | SSIM               | LPIPS  |
> | TripoSR [1]      | 19.85  | 0.753              | 0.265  | 17.68  | 0.745              | 0.277  |
> | LGM [2]          | 18.52  | 0.713              | 0.349  | 14.75  | 0.646              | 0.455  |
> | InstantMesh [3]  | 20.89  | 0.775              | 0.218  | 17.61  | 0.742              | 0.275  |
> | MeshFormer [4]   | 21.47  | 0.793              | 0.201  | 18.14  | 0.755              | 0.266  |
> | Ours             | 21.31  | 0.794              | 0.192  | 18.10  | 0.759              | 0.257  |
>
> Our method demonstrates a significant advantage over TripoSR, LGM, and InstantMesh across all three metrics on both datasets. Additionally, it is comparable with MeshFormer, with slightly lower PSNR and better SSIM and LPIPS. Note that MeshFormer is specifically designed for the single-image-to-3D task, it requires multi-view consistent normals as input during its reconstruction step. This makes it unsuitable for handling sparse-view reconstruction scenarios using captured images, such as setups with OpenIllumination, where only RGB images with camera poses are available. In contrast, our method can well handle this as shown in Table 6.
>
> We will also expand the discussion on recent works in the related work section and include the new results in the revised version.
>
> **Technical novelty (W3):**
> The novelty of our paper extends beyond simply combining LRM with differentiable iso-extraction. We also introduce a simplified version of the LRM architecture, which, as noted by Reviewer apsR, is “important and directly applicable to other LRM-based approaches.” We demonstrate that a straightforward transformer can be highly effective for 3D reconstruction, providing a more accessible path to building and scaling LRMs.
>
> Training 3D LRMs presents significantly more challenges compared to per-scene optimization (e.g., Magic3D). Techniques like differentiable rasterization, when applied in this context, often face new issues such as unstable training, necessitating novel solutions. Our carefully designed training strategies and loss functions are critical for achieving stable training and high performance. We hope our findings will benefit the community in developing scalable 3D reconstruction models.
>
> Furthermore, compared to concurrent work like InstantMesh, our method achieves superior quality as shown in the comparison table provided above.
>
>
> **Training stability (W3):**
> The limitation of directly training on mesh representations is not only due to training stability, but also because mesh representations are more prone to getting stuck in local minima during optimization. In contrast, neural radiance fields are a better choice as they can handle topological changes in a smooth and continuous manner, as noted in studies like Magic3D [6] and NeuManifold [7]. Our experiments also confirm this: single-stage training (i.e., without NeRF initialization) fails to produce reasonable results.
>
> [1] Tochilkin, Dmitry, et al. "Triposr: Fast 3d object reconstruction from a single image." arXiv (2024).
>
> [2] Tang, Jiaxiang, et al. "Lgm: Large multi-view gaussian model for high-resolution 3d content creation." ECCV, 2025.
>
> [3] Xu, Jiale, et al. "Instantmesh: Efficient 3d mesh generation from a single image with sparse-view large reconstruction models." arXiv  (2024).
>
> [4] Liu, Minghua, et al. "Meshformer: High-quality mesh generation with 3d-guided reconstruction model." NeurIPS (2024).
>
> [5] Shi, Ruoxi, et al. "Zero123++: a single image to consistent multi-view diffusion base model." arXiv (2023).
>
> [6] Lin, Chen-Hsuan, et al. "Magic3d: High-resolution text-to-3d content creation." CVPR. 2023.
>
> [7] Wei, Xinyue, et al. "Neumanifold: Neural watertight manifold reconstruction with efficient and high-quality rendering support." arXiv (2023).

---

> > ### Comment · Reviewer_ftqh · 2024-11-28
> > **Response to authors**
> >
> > Thank you for your detailed response. After thoroughly reviewing all reviews and responses, I maintain my original assessment. While the paper seems to be technically sound for me, its limited novelty remains a primary concern. The work makes notable contributions in simplifying LRM architecture through the removal of DINO image encoders and usage of sequential self-attention blocks, while demonstrating performance advantages over the similar concurrent work InstantMesh. However, the paper does not present sufficiently novel technical insights or methodological innovations to warrant a higher rating. Additionally, the lack of public availability raises concerns about reproducibility and potential impact on the 3D AIGC research community.

---

> > > ### Author Response · Authors · 2024-12-02
> > >
> > > Thanks for your feedback!
> > >
> > > We appreciate your recognition of the contributions made by our model architecture, particularly the removal of the per-view DINO encoder and the adoption of solely self-attention (across all multi-view image and triplane tokens). These design choices simplify the LRM framework while also improving quality. Notably, a similar critical transition has occurred in the NLP community—from encoder-decoder style transformers with cross-attention (as in the original "Attention Is All You Need") to decoder-only transformers with pure self-attention (as in GPT and other LLMs)—resulting in significantly improved scalability and inference quality. Our work demonstrates that such a transition (from cross-attention to self-attention, and from encoder-decoder to decoder-only architectures) can also benefit 3D large models, enhancing both scalability and quality. We hope these findings will provide valuable insights for the 3D vision and AIGC research communities in the development of future 3D large models.
> > >
> > > Additionally, we believe our simplified model is easier to implement compared to earlier LRMs. To support reproducibility, we have detailed our multi-stage training strategy, novel losses, and hyper-parameters in the paper. To further enhance accessibility, we will release a public demo of our method for the community to explore.

---

### Official Review · Reviewer_6QBv · 2024-11-02

**Soundness:** 4
**Presentation:** 3
**Contribution:** 2
**Rating:** 6
**Confidence:** 4

**Summary:**

The paper introduces MeshLRM that outputs a density and color field suitable for mesh extraction using marching cubes. Building on LRM, MeshLRM incorporates a second training stage where NeRF rendering is replaced with differentiable marching cubes (diffMC) rendering. The authors also present comprehensive ablation studies on additional losses, such as Ray Opacity Loss, which enhance the robustness of the second training stage.

**Strengths:**

The paper is clearly written, and the quality of the result is noticeably better than previous NeRF-based approach – Inst-LRM in terms of extracted mesh quality. From the perspective of engineering and practicality, the design choice proposed in this paper is sound and seems effective. The ablation in the paper is extensive and seems sound to me.

**Weaknesses:**

First, I have some reservations about the title "MeshLRM" and certain descriptions in the introduction, which may be over-claimed.  For instance, in lines 67-68, the statement, "We propose to address this with MeshLRM, a novel transformer-based large reconstruction model, designed to directly output high-fidelity 3D meshes from sparse-view inputs," could be misleading. In my view, the network does not "directly" output a mesh; instead, it produces density and color fields similar to its baseline model, which still require marching cubes to extract the mesh. The phrase "direct output mesh" more accurately describes a different family of models, such as MeshGPT [1].

2nd, another concern is the novelty of the paper. While combining LRM with diffMC is an effective and practical engineering approach, the concept appears somewhat ad hoc, mainly merging existing components rather than presenting deep or non-trivial new insights. However, this does not constitute a significant reason for rejection, as the paper still meets the typical standards of previously accepted work.

[1] MeshGPT: Generating Triangle Meshes with Decoder-Only Transformers, CVPR 24.

**Questions:**

Regarding the second stage of training, I am curious if the authors could elaborate on which parts of the network require fine-tuning. Specifically, could the model be effective with fine-tuning only the tiny MLPs or merely the last few layers of the transformer blocks?

---

> ### Author Response · Authors · 2024-11-17
>
> Abbreviations in the answers: W: Weaknesses; Q: Questions
>
> **Statement about “directly output meshes” (W1):**
> Our method is indeed different from MeshGPT, which directly predicts triangles. However, our approach incorporates marching cubes within its pipeline and employs an explicit mesh representation during training. Therefore, we consider our method to be directly generating meshes, unlike other LRM methods that rely on implicit representations during training and require additional processing to extract a mesh. We will add further clarification to clearly distinguish our approach from MeshGPT-like methods to make the claim clearer.
>
> **Novelty of the paper (W2):**
> The novelty of the paper is not limited to combining LRM and differentiable iso-extraction, but also proposing a simplified version of LRM architecture, which is mentioned to be “important and directly applicable to other LRM-based approaches” by Reviewer apsR. We show that a simple transformer is effective for 3D reconstruction, offering an easier path to building and scaling LRMs. Moreover, combining LRM and DiffMC in an end-to-end training framework encounters new challenges (like unstable training) and requires novel solutions. Our carefully designed training strategies and losses are crucial for ensuring stable training and high performance. We hope our findings can benefit the community in building scalable 3D reconstruction models. We sincerely appreciate the reviewer’s recognition that our paper “meets the typical standards of previously accepted work”!
>
> **Ablation study on fine-tuning (Q1):**
> We experimented with fine-tuning only part of the architecture and observed that all the losses converged to higher values compared to fine-tuning the entire network. In particular, the ringing artifacts on the geometry were still difficult to eliminate. Therefore, we choose to fine-tune the entire pipeline.

---

### Official Review · Reviewer_apsR · 2024-11-04

**Soundness:** 3
**Presentation:** 3
**Contribution:** 3
**Rating:** 8
**Confidence:** 4

**Summary:**

The paper presents a transformer-based framework for efficient 3D mesh reconstruction from sparse views. Unlike previous approaches relying solely on NeRF-based volumetric models with marching cubes as a post-hoc mesh extraction solution, MeshLRM incorporates differentiable mesh extraction and rendering directly within the LRM framework. The model undergoes sequential low-to-high-resolution training, followed by DiffMC-aware fine-tuning. Key contributions include DiffMC integration for improved mesh quality, a novel ray opacity loss to eliminate floaters, and training strategies like progressive resolution scaling for improved convergence.

**Strengths:**

The paper contains valuable insights that improve surface reconstruction quality and speed for LRMs. It makes a straightforward yet impactful contribution by integrating a differentiable isosurface extraction method — specifically DiffMC — into the training framework, enabling more precise mesh extraction. The proposed ray opacity loss demonstrates substantial qualitative improvements, effectively resolving the floater issue. Additionally, the insight about MLP over-parameterization leading to rendering speed slowdowns is important and directly applicable to other LRM-based approaches.

The paper is well-written and structured, clearly guiding the reader through model design, training strategy, and key components such as ray opacity loss and progressive resolution scaling, with their necessity well-supported by ablations.

**Weaknesses:**

The paper offers a specific training recipe for LRMs, but key components of this approach appear underexplored:

- The ray opacity loss is not entirely novel; ray regularization for zero density in empty space is a standard technique in NeRF approaches. While the formulation may be unique, it is unclear how it performs against simpler density-to-zero regularization or background mask losses.
- The inclusion of DiffMC, while valuable, also lacks thorough evaluation. Alternative isosurface extraction methods, such as DMTet, are available, yet the rationale for choosing DiffMC over these options is not discussed.
- The paper’s comparative analysis is limited and primarily focuses on a single baseline (Instant3D-LRM). Although MeshLRM demonstrates quantitative improvements, qualitative results are mixed, showing reduced high-frequency details in geometry and texture. For example, textures in most samples in Figure 5 lack sharpness compared to the baseline, and the geometry generally appears to be over-smoothed, e.g. frog’s legs merge with the stand in Sample 4, Fig. 5.
- The paper does not discuss recent works, such as InstantMesh or TripoSR, suggesting it may not have been updated since its initial publication on Arxiv six months before the submission. Additionally, the baselines used for comparison are approximately a year old, which dilutes the paper's relevance to the community given the rapid advancements in the field.

While the addition of isosurface extraction is important, it lacks thorough investigation. Most training strategy improvements are supported by ablations, but these are not evaluated against alternative options. The value of the paper's proposed recipe is further diluted by the existence of other potentially superior reconstruction methods, such as InstantMesh and TripoSR.

**Questions:**

- Is there evidence supporting DiffMC as a superior differentiable isosurface extraction method for mesh extraction compared to alternatives like DMTet or FlexiCubes?
- Is there comparative evidence showing that the proposed ray opacity loss formula outperforms existing density regularization techniques?
- How was the density threshold determined for DiffMC vs. standard Marching Cubes in the ablations? This choice could significantly impact surface quality, particularly in the non-finetuned case.
- Given the claimed speed improvements, how does MeshLRM's performance compare to other recently published approaches?

---

> ### Author Response · Authors · 2024-11-17
>
> Abbreviations in the answers: W: Weaknesses; Q: Questions
>
> **Ray opacity loss v.s. other regularization (W1, Q2):**
> The main difference between our ray opacity loss and previous approaches lies in the selection of supervision points. Standard density regularization uniformly supervises all points in space, including those on the mesh surface as well as in empty regions, which can lead to gradients that are less effective at guiding the model toward a clean surface. On the other hand, standard depth loss only supervises points on the mesh surface; once floaters appear during training, its gradients can move these floaters but cannot easily eliminate them. In contrast to these standard losses, our opacity loss supervises the entire space between the surface and the camera, effectively preventing floaters and stabilizing the training process. We will add a figure in the revised paper to further illustrate the difference.
>
> **Evidence of choosing DiffMC (W2, Q1):**
> Our pipeline is compatible with DMTet and Flexicubes, not limited to DiffMC. However, we chose DiffMC due to its superior processing speed and minimal GPU VRAM consumption (as detailed in the comparison table on its GitHub repository: https://github.com/SarahWeiii/diso?tab=readme-ov-file#speed-comparison). This efficiency is particularly beneficial for large-scale training, where both speed and GPU optimization are crucial.
>
>
> **Performance compared to recently published baselines (W3, W4, Q4):**
> We have added a new table to quantitatively compare our method against recent baselines. Given that input settings differ among the baselines, we follow the setup in the concurrent work MeshFormer [4] to evaluate all methods under a unified single-view-to-3D setting. Specifically, for InstantMesh, MeshFormer, and our method, we first use Zero123++ [5] to convert the input single-view image into multi-view images before performing 3D reconstruction. The other baselines follow their original configurations, taking a single-view image directly as input.
>
> The results, where all methods generate a mesh as the final output, are presented below.
>
> | Method           |        | GSO                |        |        | Omni3D             |        |
> |------------------|--------|--------------------|--------|--------|--------------------|--------|
> |                  | PSNR   | SSIM               | LPIPS  | PSNR   | SSIM               | LPIPS  |
> | TripoSR [1]      | 19.85  | 0.753              | 0.265  | 17.68  | 0.745              | 0.277  |
> | LGM [2]          | 18.52  | 0.713              | 0.349  | 14.75  | 0.646              | 0.455  |
> | InstantMesh [3]  | 20.89  | 0.775              | 0.218  | 17.61  | 0.742              | 0.275  |
> | MeshFormer [4]   | 21.47  | 0.793              | 0.201  | 18.14  | 0.755              | 0.266  |
> | Ours             | 21.31  | 0.794              | 0.192  | 18.10  | 0.759              | 0.257  |
>
>
> Our method demonstrates a significant advantage over TripoSR, LGM, and InstantMesh across all three metrics on both datasets. Additionally, it is comparable with MeshFormer, with slightly lower PSNR and better SSIM and LPIPS. Note that MeshFormer is specifically designed for the single-image-to-3D task, it requires multi-view consistent normals as input during its reconstruction step. This makes it unsuitable for handling sparse-view reconstruction scenarios using captured images, such as setups with OpenIllumination, where only RGB images with camera poses are available. In contrast, our method can well handle this as shown in Table 6.
>
> We will also expand the discussion on recent works in the related work section and include the new results in the revised version.
>
>
> **Density threshold (Q3):**
> Both DiffMC and standard Marching Cubes use the same density threshold, which is selected based on the overall quality of the Marching Cube mesh. During stage two fine-tuning, DiffMC maintains this threshold. In general, we observe that, without DiffMC fine-tuning, the Marching Cubes artifacts shown in the paper are unavoidable with various thresholds. While carefully tuning the threshold for a single scene can reduce the artifacts to some extent, the optimal threshold differs notably across scenes. Our DiffMC fine-tuning effectively resolves these issues, allowing for high-quality mesh reconstruction with the consistent pre-selected threshold.
>
>
> [1] Tochilkin, Dmitry, et al. "Triposr: Fast 3d object reconstruction from a single image." arXiv (2024).
>
> [2] Tang, Jiaxiang, et al. "Lgm: Large multi-view gaussian model for high-resolution 3d content creation." ECCV, 2025.
>
> [3] Xu, Jiale, et al. "Instantmesh: Efficient 3d mesh generation from a single image with sparse-view large reconstruction models." arXiv (2024).
>
> [4] Liu, Minghua, et al. "Meshformer: High-quality mesh generation with 3d-guided reconstruction model." NeurIPS (2024).
>
> [5] Shi, Ruoxi, et al. "Zero123++: a single image to consistent multi-view diffusion base model." arXiv (2023).

---

> > ### Comment · Reviewer_apsR · 2024-11-20
> >
> > Thank you for the clarification! Your response addresses my concerns, and I am willing to improve my scores, especially since multiple concerns from other reviewers have also been addressed

---

> > > ### Author Response · Authors · 2024-11-20
> > >
> > > We are glad to hear that all your concerns have been addressed and thank you for raising the scores!

---

### Author Response · Authors · 2024-11-24
**Revised paper**

Hi everyone,

Thank you for your valuable feedback. We have revised the paper based on your suggestions and highlighted the changes in blue for your convenience. The updated version has been uploaded—please feel free to review it.

Best regards,

The Authors

---

### Meta-Review · Area_Chair_ZbbC · 2024-12-21

**Metareview:**

The paper received mixed reviews, with one accept, two borderline accepts, and one borderline reject. While Reviewer apsR expressed a generally positive view of the paper, all reviewers raised concerns regarding the novelty of the work. The rebuttal emphasized the paper’s contribution as a simplified version of the LRM architecture. However, this argument was not sufficiently convincing to change the reviewers’ perspectives, particularly those of Reviewer ftqh and Reviewer 1jbs.

Reviewer apsR highlighted that the ray opacity loss introduced in the paper is not entirely novel, and ray regularization for zero density in empty space is a standard technique in NeRF approaches. Reviewer 6QBv criticized the method for simply combining LRM with diffMC, describing the approach as somewhat ad hoc (while Reviewer 6QBv did not take this as a significant reason for rejection, the Area Chair agrees with Reviewer 1jbs that 3D generation should not be held to a lower standard for acceptance). Reviewer ftqh noted that earlier works, such as Magic3D and InstantMesh, have already explored two-stage 3D generation pipelines by combining NeRF and mesh representations. While the usage of mesh representations is reasonable, it is not novel in this context. Reviewer ftqh also raised a significant concern regarding the lack of publicly available code, which is crucial for reproducibility and further research by the community. Reviewer 1jbs agreed with the other reviewers on the novelty issue and actively participated in the discussion, demonstrating a strong sense of responsibility in their role as a reviewer. Reviewer 1jbs also highlighted, alongside Reviewer ftqh, the lack of public code availability, noting that the authors only plan to provide a demo in the future. This limitation significantly undermines the impact of this work which is mainly an engineering contribution.

The area chair has thoroughly reviewed the paper, supplementary materials, and the discussions among reviewers. The paper’s lack of novelty significantly limits its contribution to the field, as the proposed method largely combines existing techniques without introducing substantial new insights. While providing open source code is also commendable for impactful engineering contributions, the paper falls short in this regard. The active participation of Reviewer 1jbs in the discussion is particularly valued, as it demonstrated a thoughtful and responsible evaluation of the work, ultimately leaning toward a weak reject. The concerns of Reviewer 1jbs, alongside those of other reviewers, highlight that the technical contributions do not meet the high standards of ICLR.

**Additional Comments On Reviewer Discussion:**

The authors’ rebuttal emphasizes the contribution of a simplified version of the LRM architecture, but this argument did not sufficiently address the concerns raised by the reviewers. Specifically, Reviewer ftqh and Reviewer 1jbs expressed doubts about the technical contribution of the approach, which remains a major concern despite the rebuttal. Additionally, Reviewer ftqh raised an important issue regarding the availability of code, which the rebuttal did not fully resolve. This issue was also acknowledged by Reviewer 1jbs, who has actively participated in the discussion.

---

### Decision · Program_Chairs · 2025-01-22

Reject